

# Estimating species sensitivity distributions on the basis of readily obtainable descriptors and toxicity data for three species of algae, crustaceans, and fish

Yuichi Iwasaki and  Kiyan Sorgog

Research Institute of Science for Safety and Sustainability, National Institute of Advanced Industrial Science and Technology, Tsukuba, Ibaraki, Japan

## ABSTRACT

Estimation of species sensitivity distributions (SSDs) is a crucial approach to predicting ecological risks and water quality benchmarks, but the amount of data required to implement this approach is a serious constraint on the application of SSDs to chemicals for which there are few or no toxicity data. The development of statistical models to directly estimate the mean and standard deviation (SD) of the logarithms of log-normally distributed SSDs has recently been proposed to overcome this problem. To predict these two parameters, we developed multiple linear regression models that included, in addition to readily obtainable descriptors, the mean and SD of the logarithms of the concentrations that are acutely toxic to one algal, one crustacean, and one fish species, as predictors. We hypothesized that use of the three species' mean and SD would improve the accuracy of the predicted means and SDs of the logarithms of the SSDs. We derived SSDs for 60 chemicals based on quality-assured acute toxicity data. Forty-five of the chemicals were used for model fitting, and 15 for external validation. Our results supported previous findings that models developed on the basis of only descriptors such as $\log K_{OW}$ had limited ability to predict the mean and SD of SSD (e.g., $r^2 = 0.62$ and 0.49, respectively). Inclusion of the three species' mean and SD, in addition to the descriptors, in the models markedly improved the predictions of the means and SDs of SSDs (e.g., $r^2 = 0.96$ and 0.75, respectively). We conclude that use of the three species' mean and SD is promising for more accurately estimating an SSD and thus the hazardous concentration for 5% of species in cases where limited ecotoxicity data are available.

## INTRODUCTION

In the ecological risk assessment of chemicals, the use of species sensitivity distributions (SSDs) is fundamental to deriving "safe" concentrations such as the predicted no effect concentration (PNEC) and water quality benchmarks, as well as the potentially affected

Corresponding author
Yuichi Iwasaki, yuichiwsk@gmail.com

percentages of species (*Vaal et al., 1997*; *Posthuma, Suter & Traas, 2002*; *Iwasaki et al., 2015*; *Belanger et al., 2017*; *Carr & Belanger, 2019*; *Posthuma et al., 2019*; *Sorgog & Kamo, 2019*). SSDs can be derived on the basis of acute or chronic toxicity data, or both. To derive an SSD for a given chemical, a set of toxicity data is fitted to statistical distributions such as a log-normal or log-logistic distribution. Therefore, toxicity results for multiple species are required, and the minimum sample sizes (i.e., the minimum number of species required) typically vary between five and 10, depending on the regulatory jurisdiction (*Belanger et al., 2017*). This data-demanding feature is a critical barrier to the application of SSD to chemicals for which there are few, or no, toxicity data (*Hoondert et al., 2019*; *Hiki & Iwasaki, 2020*).

To address this data-demand issue, the development of statistical models with physicochemical descriptors to estimate the mean and standard deviation (SD) of the logarithms of log-normally distributed SSDs has recently been proposed (*Hoondert et al., 2019*). Development of separate models for the mean and SD of the logarithms of SSDs rather than models that estimate the hazardous concentration for X% of species (HC$_x$; typically, HC$_5$) based on an SSD is valuable for two different reasons. First, both of these parameters are required to estimate the fraction of species affected by a given concentration of a toxic substance as well as the concentration that is hazardous for a given percent of the species (*Posthuma, Suter & Traas, 2002*). Second, the mean and SD have inherently different statistical properties (e.g., the SD of the logarithms of the SSD is related to inter-species differences in sensitivity). Although the results are preliminary, the coefficients of determination ($R^2$) of models developed for the SDs of the logarithms of the SSDs were less than 0.4 (*Hoondert et al., 2019*). However, the parameters that characterize SSDs can be more robust than the concentrations that are acutely toxic to individual species (*Iwasaki et al., 2015*). A modeling approach aimed at characterizing SSDs therefore seemed worth pursuing.

The major concept that we tested here was to make use of the mean and SD of toxicity values for three biological species chosen (randomly) from three biological groups (algae, crustaceans, and fish), in addition to readily obtainable descriptors, to improve the model prediction of SSDs. This idea is similar to the application of QAAR (quantitative activity–activity relationship) and QSAAR (quantitative structure–activity–activity relationship) models (*Furuhama, Hayashi & Yamamoto, 2019*). Acquiring at least one set of toxicity data from each of algae, crustaceans, and fish may not be feasible for many chemicals, but these biological groups are often the most representative in terms of data availability (*Aurisano et al., 2019*). Given the difficulties in predicting SSD parameters on the basis of readily obtainable descriptors alone (*Hoondert et al., 2019*), an approach that lies in between prediction based on "only descriptor-based" models and prediction based only on experimentally obtained toxicity data could be promising.

Here, acute toxicity data were used to estimate SSDs because of the paucity of quality-assured acute toxicity data. Our specific aims were threefold (Fig. 1). We first aimed to develop and test linear regression models with readily obtainable descriptors, such as the log$_{10}$-transformed octanol–water partition coefficient (log $K_{OW}$), to predict the means and SDs of acute log-normal SSDs. By doing this, we evaluated the previous conclusions

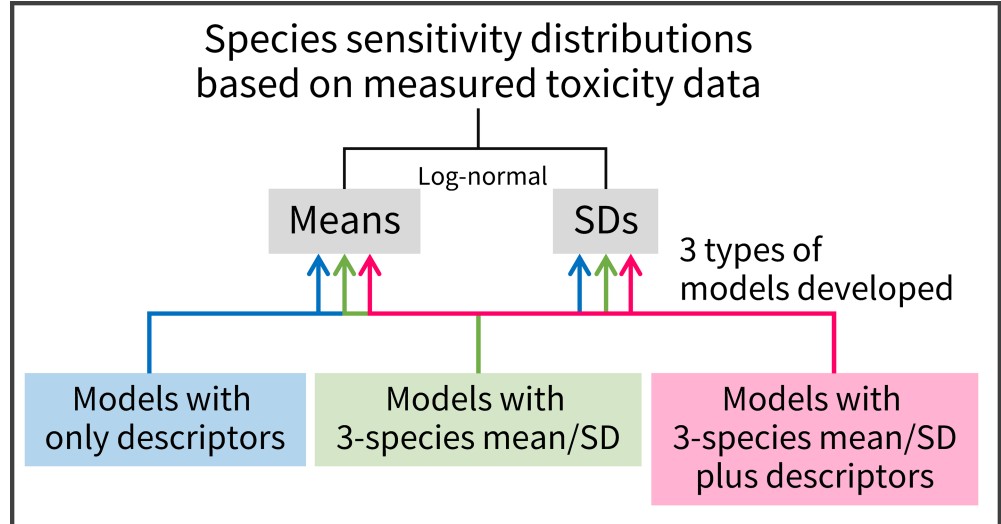

**Figure 1** **A summary schematic of model development in the present study.** Three types of predictive models were used to estimate means and standard deviations (SDs) of the logarithms of log-normally distributed species sensitivity distributions based on measured toxicity data.

of *Hoondert et al. (2019)* about model performance. Second, we aimed to develop linear regression models with predictors of means and SDs estimated on the basis of toxicity values for three species chosen from three biological groups (algae, crustaceans, and fish). Finally, we aimed to develop and test linear regression models with predictors of means and SDs based on the three species' toxicity data, as well as the descriptors, to predict the means and SDs of SSDs. Although several studies have derived SSDs based on three species' toxicity data and have assessed the associated uncertainties of $HC_{50}$ and $HC_5$, particularly in the field of life-cycle impact assessment (*Van Zelm et al., 2007*; *Golsteijn et al., 2012*; *Douziech et al., 2020*), to our knowledge no attempt has been made to develop and test the models by the combined use of the two different types of predictors (i.e., descriptors and three species' mean and SD).

## MATERIALS AND METHODS

### Data collection

All acute toxicity data (e.g., the 50% effect concentration [$EC_{50}$] and median lethal concentration [$LC_{50}$]) were collected from datasets used for initial environmental risk assessments (IERAs) of chemicals; these have been performed annually from 2002 by the Ministry of Environment (MoE), Japan (*MoE Japan, 2018*). We obtained some of the datasets (2002–2014) from the National Institute of Advanced Industrial Science and Technology (AIST)-Multi-purpose Ecological Risk Assessment and Management Tool (AIST-MeRAM; Copyright No: H28PRO-2007; *Lin et al., 2020*), and we obtained the more recent datasets from original pdf documents (*MoE Japan, 2018*). In IERAs, the quality of the toxicity data collected is examined by referring to international or domestic test guidelines or by evaluating the test conditions, test species, and physicochemical properties
of the chemical being tested. The quality is assigned to one of four categories (Reliable without restriction, Reliable with restrictions, Not reliable, and Not assignable) in accord with *OECD (2007)* and *Klimisch, Andreae & Tillmann (1997)*. Here, we used the toxicity data assigned to the former two categories, which were also used to derive PNECs in the original IERAs, to minimize undesired variability in toxicity data and thereby to accurately estimate the means and SDs of SSDs. The acute toxicity measures analyzed were dominated by $EC_{50}$s and $LC_{50}$s, but they also included other similar measures (e.g., median tolerance limit ($TL_m$) and 50% inhibitory concentration ($IC_{50}$)). To estimate the SSD of a chemical, if multiple toxicity data were available for a single species, the calculated geometric mean was used (*Iwasaki et al., 2015*; *Hiki & Iwasaki, 2020*).

To estimate log-normally distributed SSDs, we also used the following criteria: (1) toxicity data for $\geq 8$ species, including at least one species from each of the three biological groups (algae, crustaceans, and fish) were available; (2) the assumption of normality of SSD was not rejected by the Shapiro–Wilk test ($\alpha = 0.05$ with Holm's $P$-value adjustment; *Hiki & Iwasaki (2020)*); and (3) all the information about the predictors below was acquired, excluding that for inorganic substances. Application of these criteria yielded a total of 60 chemicals to be analyzed (see Table S1 in the Supplementary File for the list of chemicals). Use of a relatively large numbers of species (10–55) has been advocated for accurate estimation of SSDs (*Newman et al., 2000*; *Carr & Belanger, 2019*; *Hiki & Iwasaki, 2020*), but a relatively small numbers of species (<10) has been used previously, and a minimum sample size of 5 has been suggested (*TenBrook et al., 2009*). However, if 10 species had been used as the threshold in the present study, the number of chemicals analyzed would have been reduced by almost a factor of two. We therefore used a minimum sample size of 8 species as a tradeoff between the number of chemicals analyzed and the uncertainties in the estimated SSDs. We were able to estimate 29 SSDs with data on 8 or 9 species, and we could estimate 31 SSDs with information for at least 10 species (see Table S1 in the Supplementary File for more details).

The means and SDs of the logarithms of the log-normally distributed SSDs for 60 chemicals were calculated based on full datasets. The means and SDs were obtained by calculating the means and sample SDs of the $log_{10}$-transformed toxicity data. A sample SD was equated to the square root of the sum of the squared deviations from the mean divided by n $-1$, where n was the number of species. In the present study, we have referred to the statistics that we calculated as SSD means and SDs, although they were actually the means and SDs of the $log_{10}$-transformed values. Those means and SDs were used as objective variables in the later regression analysis. The $log_{10}$-transformed values of $HC_5$ were derived by using Eq. (1):

$$HC_5 = mean - 1.645 \times SD. \tag{1}$$

The 95% confidence intervals of the $log_{10}$-transformed $HC_5$s based on SSDs derived from measured toxicity values were also estimated according to *Aldenberg, Jaworska & Traas (2002)* (see *Sorgog & Kamo, 2019* for the R code). As with SSD means and SDs, we hereafter referred to $log_{10}$-transformed $HC_5$s as $HC_5$s.

## Predictors

Molecular weight (MW) obtained from the IERA reports, as well as log $K_{OW}$ (estimated on the basis of KOWWIN v1.69), $\log_{10}$-transformed vapor pressure at 25 °C (units: mm Hg; MPBPWIN v1.43; hereafter, log VP), and the calculated rating value for the time required for primary biodegradation (BIOWIN4; BIOWIN v4.10)—all of which were estimated by using the EPI (Estimation Programs Interface) Suite v4.11 (*US EPA, 2017*)—were used as predictors in the regression analyses. These predictors were selected on the basis of the results of *Hoondert et al. (2019)* and *Iwasaki & Hayashi (2020)*. Water solubility was not used, because it was estimated from log $K_{OW}$ (*US EPA, 2017*) and was highly correlated with log $K_{OW}$ ($r = -0.96$). As predictors for the regression analyses, we also used two binary variables (Class 3 and Class 4) concerning whether a given chemical was categorized as Class 3 (reactive chemicals) or Class 4 (specifically acting chemicals) on the basis of the Verhaar mode-of-action classification (*Verhaar, Van Leeuwen & Hermens, 1992*), as estimated by using the OECD QSAR Toolbox (v4.4.1; https://qsartoolbox.org/). This is because accumulated evidence indicates that the means and SDs of SSDs for Class 3 and 4 chemicals differ from those of other chemicals (*Vaal et al., 1997*; *Hendriks et al., 2013*; *Hiki & Iwasaki, 2020*). Because the influences of chemical classes other than Class 3 and Class 4 were not evident in those studies, we chose not to include a categorical variable representing the Verhaar mode-of-action classification to save the degree of freedom. Of the 60 chemicals that we studied, 27% were assigned to Class 1 (narcosis or baseline toxicity), 8% to Class 2 (less inert compounds), 37% to Class 3, 12% to Class 4, and 17% to Class 5 (unclassified; see Table S1 for the raw data).

For each chemical, we also calculated the mean and SD of the toxicity data for an alga, crustacean, and fish and hereafter refer to these 3-species-based predictors as the 3-species mean and SD, respectively. For purposes of this calculation, we first produced 3000 bootstrapped sample sets of 3-species means and SDs for each chemical by randomly choosing three species that had been tested (i.e., choosing one species from each of the three biological groups by non-parametric bootstrapping). The median value of the 3-species means and the corresponding SD were then selected for each chemical, and these were used as predictors based on the 3-species data in the later analysis. Note that the three species adopted here were also included to calculate the means and SDs of SSDs, which were used as objective variables in the regression analysis. The use of the 3-species mean and SD as predictors in the regression analysis is unconventional, because the toxicity data used for calculating the predictors were also utilized to calculate objective variables. However, it should be plausible given the limited availability of toxicity data and our focus on examining the utility of the predictors in estimating SSDs.

## Model evaluation

All statistical analyses were performed by using R 3.6.3 (*R Core Team, 2020*). All the associated data and the R code are available in the Supplementary File. For model development and evaluation, 45 chemicals were randomly chosen from the 60 chemicals and used as training data, and the remaining chemicals were used as test data for out-of-bag model evaluation. The median values and ranges of SSD means and SDs and the descriptors

**Table 1** Summary information on the training and test data used for model development and evaluation.

| Variable | Training data | Test data |
|---|---|---|
| Number of chemicals | 45 | 15 |
| Number of Class 3 chemicals | 15 | 7 |
| Number of Class 4 chemicals | 6 | 1 |
| Number of species[*] | 10 (8–19) | 9 (8–21) |
| SSD mean[*] | 0.92 (−2.71–4.50) | 1.06 (−1.05–4.40) |
| SSD SD[*] | 0.67 (0.25–1.65) | 0.42 (0.26–1.39) |
| Molecular weight[*] | 137 (56–391) | 144 (61–221) |
| Log $K_{OW}$[*] | 2.65 (−1.47–7.43) | 2.18 (−1.61–3.93) |
| Log VP[*] | −1.54 (−6.56–2.42) | −0.91 (−4.88–1.31) |
| Biodegradability[*] | 3.56 (2.85–4.74) | 3.58 (3.09–3.92) |

**Notes.**

[*]Median and range (minimum and maximum) are shown.

SSD, species sensitivity distribution; Class 3 or 4, reactive chemicals or specifically acting chemicals, respectively, based on the Verhaar mode of action classification (*Verhaar, Van Leeuwen & Hermens, 1992*); SD, standard deviation; log $K_{OW}$, $\log_{10}$-transformed octanolwater partition coefficient; log VP, $\log_{10}$-transformed vapor pressure at 25 °C (see 'Materials and Methods' for further details).

of the test data were generally within those of the training data, although the minimum value of log $K_{OW}$ in the test data was slightly lower (Table 1).

To predict the means and SDs of the SSDs, we developed three types of the two linear regression models, namely: (1) the best models with descriptors only; (2) the models with the 3-species-based predictors only; and (3) the best models with both descriptors and 3-species-based predictors. For the development of models (2) and (3), we used the 3-species mean and SD to predict the means and SDs of the SSDs, respectively. The 3-species mean (or SD) was therefore not used to predict the SSD SDs (or means). Also, for model developments (1) and (3), models were selected on the basis of the corrected Akaike information criterion (AICc; *Burnham & Anderson, 2002*; *Burnham, Anderson & Huyvaert, 2011*) with the training data, by using the "dredge" function in the R package "MuMIn." The best models with the smallest AICc values were selected among all possible models considered. Although our major focus was on selecting the best models for parsimonious predictions, full-model-averaged standardized parameter estimates were calculated by using the "model.avg" function of "MuMIn" to examine the relative variable importance (*Galipaud, Gillingham & Dechaume-Moncharmont, 2017*). We also performed complementary analyses by fitting the best models with both descriptors and 3-species-based predictors to evaluate the robustness of the model selection results based on the median estimates. In the analyses, instead of the median values of 3-species means and SDs, we used five sets of 3-species means and SDs. The five sets were randomly chosen from the 3000 bootstrapped samples, and we tested the hypothesis that the results of model selection and fitting were unchanged by use of the five sets.

By referring to *OECD (2017)* and *Furuhama, Hayashi & Yamamoto (2018)*, as internal validation we calculated two commonly used statistics—the coefficient of determination ($r^2$) and the root mean square error (RMSE)—to evaluate the goodness of fit of individual models, and we determined the leave-one-out cross-validated coefficient of determination

($q^2$) to evaluate the model robustness by using the R package "caret" (*Kuhn, 2008*). In addition, as external validation, we determined the RMSE (RMSE$_{ext}$) of the test data, the squared correlation coefficient between the measured and predicted values ($r^2_{ext}$), and the explained variance ($Q^2_{ext}$) of the test data. We also used the developed models to predict the means and SDs of SSDs and then the HC$_5$s, which were compared with HC$_5$s estimated on the basis of toxicity data. To evaluate the uncertainty of the HC$_5$s predicted from the developed models, the 95% confidence intervals were calculated using non-parametric bootstrapping (i.e., the 95% confidence intervals were equated to the 75th and 2925th of 3000 sorted HC$_5$s based on the best models fitted to 3000 resampled datasets).

## RESULTS

Predictors included in the best model developed with descriptors alone for SSD mean were Class 4, log $K_{OW}$, and Class 3, all of which had higher (negative) full-model-averaged standardized parameter estimates than the other variables (<−0.19; Table 2). Similarly, the best model developed with descriptors alone for SSD SD included only Class 4. However, along with Class 4, Log VP and MW had higher full-model-averaged standardized parameter estimates than the other variables (Table 2). Although the RMSE and RMSE$_{ext}$ values of these best models were both below approximately 0.8 for the SSD mean and below approximately 0.2 for the SD (i.e., both within a factor of 10 of the values based only on measured toxicity data), the $r^2$ values were low (below approximately 0.6; Table 3 and Fig. 2).

Incorporating the means or SDs of the toxicity values for three species markedly improved the performances of the models (e.g., see changes in $r^2$ values in Table 3; Fig. 2). The best models for both descriptors and 3-species-based mean/SD were (see the Supplementary File for more details):

$$\text{SSD mean} = 0.07 + 1.02 \times \text{3-species mean} - 0.27 \times \text{Class 4} \qquad (2)$$
$$\text{SSD SD} = 0.39 + 0.48 \times \text{3-species SD} + 0.30 \times \text{Class 4} + 0.02 \times \text{log VP}. \qquad (3)$$

Our finding that the differences in AICc values between models with descriptors alone and those with the 3-species mean or SD were over 20 also indicated the substantial model improvements with the latter (Table 3; also see *Burnham, Anderson & Huyvaert, 2011*; *Iwasaki & Hayashi, 2020*). Particularly for the models for SSD mean, inclusion of only the 3-species mean as a predictor increased the $r^2$ values from 0.62 to 0.96, and this was also evident from the prominently high full-model-averaged standardized parameter estimates of this predictor (i.e., 0.94; Table 2). Inclusion of only the 3-species SD also improved the model fit in the prediction of SSD SD (e.g., $r^2$ increased from 0.49 to 0.70; Table 3), and model performance was further improved by including two descriptors (Log VP and Class 4; Tables 2 and 3). The best model for predicting SSD mean had RMSE and RMSE$_{ext}$ values of <0.5, and that for predicting SSD SD had RMSE and RMSE$_{ext}$ values of <0.2 (Table 3). Moreover, when we selected and evaluated the best models by using five randomly

**Table 2  Full-model-averaged standardized parameter estimates for the four models selected for predicting the mean and standard deviation (SD) of species sensitivity distribution (SSD).**

| Predictor | Model for SSD mean | | Model for SSD SD | |
| --- | --- | --- | --- | --- |
| | Descriptors | 3-species mean + descriptors | Descriptors | 3-species SD + descriptors |
| 3-species mean | NA | 0.938 | NA | NA |
| 3-species SD | NA | NA | NA | 0.706 |
| Biodegradability | 0.009 | 0.006 | −0.057 | −0.012 |
| Class 3 | −0.191 | 0.002 | 0.004 | 0.005 |
| Class 4 | −0.580 | −0.025 | 0.687 | 0.231 |
| Log $K_{OW}$ | −0.349 | −0.003 | 0.020 | 0.005 |
| Log VP | 0.049 | 0.019 | 0.138 | 0.107 |
| Molecular weight | 0.016 | −0.018 | 0.093 | 0.012 |

**Notes.**

Higher values of full-model-averaged standardized parameter estimates indicate higher relative importance of the variable in the model that was developed. Underlined predictors are those included in the best models. NA, not applicable. See the footnote to Table 1 for further details about the predictors.

**Table 3  Model evaluation for estimating the mean and standard deviation (SD) of species sensitivity distribution (SSD).**

| Statistics | Models for SSD mean | | | Models for SSD SD | | |
| --- | --- | --- | --- | --- | --- | --- |
| | Descriptors | 3-species mean | 3-species mean + descriptors | Descriptors | 3-species SD | 3 species SD + descriptors |
| AICc | 121.7 | 20.0 | 19.5 | 5.0 | −19.3 | −22.4 |
| Internal validation | | | | | | |
| $r^2$ | 0.62 | 0.96 | 0.96 | 0.49 | 0.70 | 0.75 |
| RMSE | 0.82 | 0.28 | 0.27 | 0.24 | 0.18 | 0.17 |
| $q^2$ | 0.52 | 0.95 | 0.95 | 0.41 | 0.67 | 0.69 |
| External validation | | | | | | |
| $r^2_{ext}$ | 0.78 | 0.84 | 0.86 | 0.34 | 0.79 | 0.75 |
| RMSE$_{ext}$ | 0.58 | 0.50 | 0.47 | 0.28 | 0.16 | 0.17 |
| $Q^2_{ext}$ | 0.75 | 0.84 | 0.86 | 0.25 | 0.69 | 0.66 |
| # of parameters | 3 | 1 | 2 | 1 | 1 | 3 |

**Notes.**

AICc, corrected Akaike information criterion; $r^2$, coefficient of determination; RMSE, root mean square error; $q^2$, leave-one-out cross-validated coefficient of determination; RMSE$_{ext}$, root mean square error in the test data; $r^2_{ext}$, squared correlation coefficient between measured and predicted values; $Q^2_{ext}$, variance explained in the test data.

chosen bootstrapped samples, the statistics for model evaluation usually changed little (particularly, the coefficients of variation in the three statistics for the internal validation ($r^2$, RMSE, and $q^2$) were <6%), although the best models sometimes had one more, or one less, predictor compared with the best models developed above (see Tables S2 and S3 in the Supplementary File).

As was expected from results discussed above, the HC$_5$ values predicted from SSD means and SDs that were estimated on the basis of models with the 3-species mean/SD and the best models developed with the 3-species mean/SD plus descriptors were highly correlated with the HC$_5$ values estimated on the basis of the measured toxicity data ($r^2 \geq 0.95$; Fig. 3), unlike those based on the best models with descriptors alone ($r^2 = 0.67$).

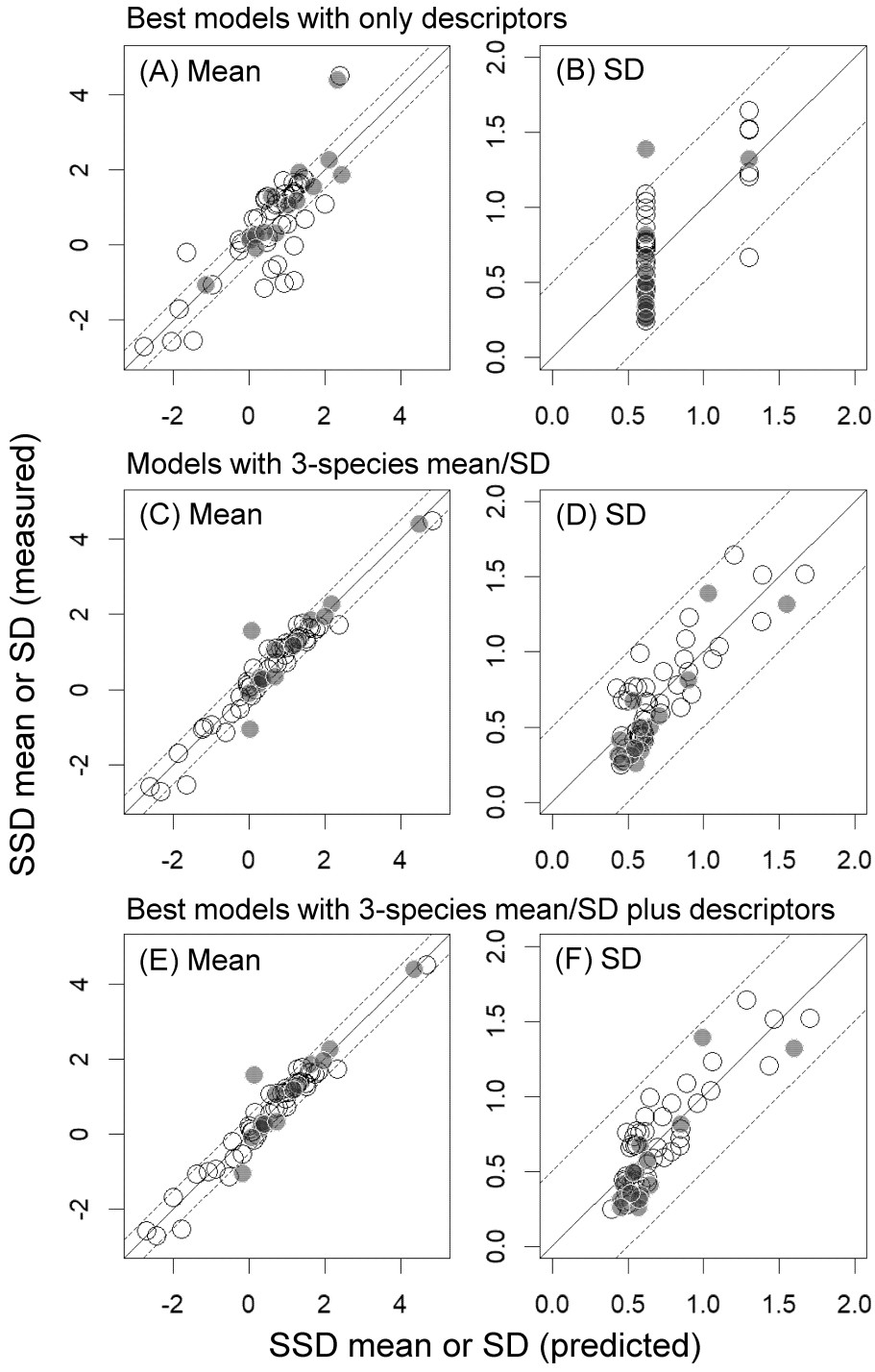

**Figure 2** **Mean and standard deviation (SD) of the logarithms of log-normally distributed species sensitivity distributions (SSDs) estimated based on measured toxicity data alone or predicted from models developed in the present study.** (A & B) Mean and SD predicted from best models developed with descriptors alone. (C & D) Mean and SD predicted from models developed with the 3-species mean/SD. (E & F) Mean and SD predicted from best models developed with the 3-species mean/SD plus descriptors. See Tables 2 and 3 for details of best models and model evaluation, respectively. White and gray dots indicate training data and test data, respectively.

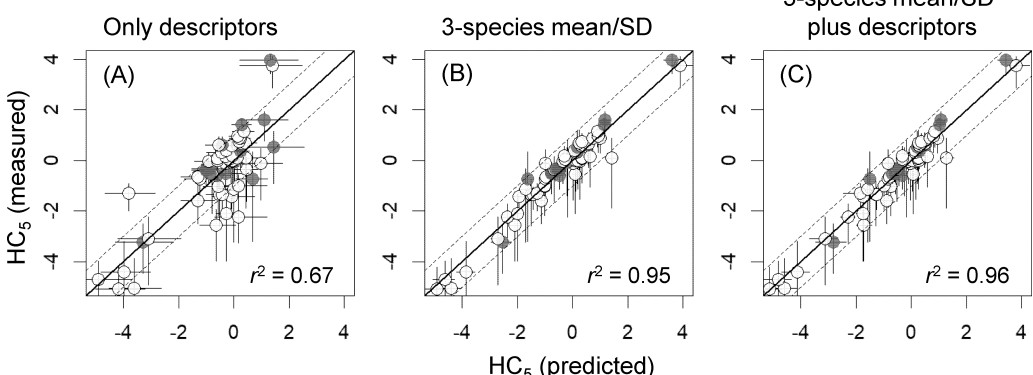

**Figure 3** Log$_{10}$-transformed hazardous concentrations for 5% of species (HC$_5$s), estimated by species sensitivity distributions based on measured toxicity data or estimated by using three types of models (A–C; see Table 2 for more information). *X*- and *y*-axis error bars respectively indicate 95% confidence intervals for HC$_5$, as estimated by non-parametric bootstrapping, and those based on measured toxicity data (see the text for the details). White and gray dots indicate training data and test data, respectively. Values of $r^2$ indicate squared correlation coefficients between the measured and predicted HC$_5$ values. SD, standard deviation.

Also, in the HC$_5$ predictions based on those models with the 3-species mean/SD, only one chemical (chloroform) was outside a factor of 10 compared with the HC$_5$ values derived from the measured data, whereas 15 chemicals were outside the range in the prediction based on models with descriptors alone (Fig. 3).

## DISCUSSION

Our results based on SSDs derived for 60 chemicals clearly showed that the models developed on the basis of descriptors alone had limited ability to predict the mean and SD of SSD and thereby HC$_5$; for example, this is particularly apparent from our finding that, when descriptors alone were used, the predicted HC$_5$ values for 25% of chemicals deviated by more than a factor of 10 from HC$_5$ values estimated on the basis of the measured toxicity data (Fig. 3). These results are generally consistent with the conclusion of *Hoondert et al. (2019)* that uncertainties associated with prediction of the mean and SD of SSDs and HC$_5$ by using descriptors may be high (see Fig. S3 of *Hoondert et al., 2019* for the HC$_5$ prediction). However, compared with our results, the variations not explained by the models were somewhat larger in the work of *Hoondert et al. (2019)*. This difference was obvious from the $r^2$ values of their best models (0.56 and 0.29, respectively; see also *Iwasaki & Hayashi, 2020*), even though the predictors that they used were similar to ours, with the exception of the two binary variables (Class 3 and Class 4). It is difficult to determine the underlying reasons for this difference in predictive ability. Two apparent differences were the quantity of data and whether the quality of each of the experimentally obtained toxicity values was examined. *Hoondert et al. (2019)* analyzed SSDs based on acute toxicity data for 558 chemicals, which was approximately 9 times the number of chemicals considered in the present study; however, they did not take the quality of each of the experimentally

obtained toxicity values into consideration although they differentiated the quality of SSDs based on the inclusion of extrapolated toxicity data. Despite the clear trade-off, both kinds of data should be valuable in developing predictive models that are well tested. There is a need, however, for further testing of the utility of 3-species predictors with large datasets.

Compared with the models with descriptors alone, the addition of a 3-species mean/SD markedly improved the predictions of the mean and SD of SSD (see Table 3), as well as of $HC_5$. These results suggest that using not only descriptors but also the mean and SD of toxicity data for three species, each of which belongs to one of the three commonly tested biological groups (algae, crustaceans, and fish), is a promising way of producing a more reliable SSD derivation in cases where limited ecotoxicity data are available. *Douziech et al. (2020)* also demonstrated that $HC_5$ values estimated on the basis of acute toxicity data for three species of algae, daphnia, and fish were mostly within a factor of 10 of $HC_5$ values based on all measured toxicity values. Although the uncertainties associated with our model estimates of $HC_5$ were relatively small—particularly for those obtained by including the 3-species mean/SD as a predictor (see the $x$-axis error bars in Fig. 3), this prediction assumed that the means and SDs of SSDs calculated from the measured toxicity data for at least 8 species were accurate. Caution is therefore required in the use of such model-based estimates in ecological risk assessments because of the inherent uncertainties in $HC_5$ values based on measured toxicity data ($y$-axis error bars in Fig. 3; see also de Zwart 2002 and *Hiki & Iwasaki, 2020*) as well as the large uncertainties in $HC_5$ values estimated on the basis of 3 species toxicity data (*Douziech et al., 2020*).

The binary predictor, Class 4 (specifically acting chemicals), was included in all four best models, with relatively large negative full-model-averaged standardized parameter estimates in the models predicting SSD means and relatively large positive full-model-averaged standardized parameter estimates in the models predicting SSD SDs (see Table 2). These results support previous empirical evidence that specifically acting chemicals are generally more toxic than other classes of chemicals, and that inter-species sensitivity to these chemicals varies more widely than that to other classes of chemicals (*Vaal et al., 1997*; *Hendriks et al., 2013*; *Hiki & Iwasaki, 2020*); they also imply that such patterns cannot be fully captured by using only the 3-species mean and SD. The variations in toxicity values within the biological groups appeared to be relatively high for Class 4 chemicals (Fig. S1), and this may have led to the importance of Class 4 in the models that we developed. However, it is difficult to test this hypothesis robustly owing to the limited number of Class 4 chemicals in our study (a total of 7 chemicals). Another binary predictor, Class 3 (reactive chemicals), was selected in the best model with descriptors alone for the SSD mean, and not in the other best models, suggesting that inclusion of the 3-species mean could play an alternative role for Class 3. The predictive abilities of the best models developed with the 3-species mean/SD plus descriptors were similar among Classes 1–5 based on the Verhaar mode-of-action classification (see Fig. S2 for the differences between the predicted and measured means and SDs of the SSDs).

Log $K_{OW}$ (or log P) is commonly used in QSAR (quantitative structure–activity relationship) models to predict ecotoxicity (*Furuhama et al., 2010*; *Mayo-Bean et al., 2017*), and it was indeed included in the best model with descriptors alone for SSD mean. Log

VP was included in the best model with the 3-species SD plus descriptors (Table 2), and its regression coefficient was positive but non-significant ($P = 0.072$). This may indicate that the increased variation in species sensitivity (i.e., the SD of SSD) was associated with the difficulty in toxicity testing of the more volatile chemicals. However, a clear positive relationship between the SD of SSD and log VP was not apparent owing to the large variation in the SD (Table S1). Note that although use of estimated values of these descriptors is efficient, particularly if the number of chemicals of concern is large, there are uncertainties associated with the estimated values (US EPA, 2007; Enoch et al., 2008) that may have affected the development of the model and the accuracy of its predictions. Further tests of the developed models with experimentally obtained values of descriptors would therefore be valuable, even though experimental results are associated with uncertainties that can vary as a function of experimental conditions.

## CONCLUSIONS

We demonstrated here that the use of measured toxicity data for three species selected from the three commonly tested biological groups (algae, crustaceans, and fish) markedly improved the model predictions for the means and SDs of log-normal SSDs based on acute toxicity. Furthermore, the $HC_5$s in SSDs predicted by using such models were generally within a factor of 10 of the values based only on measured toxicity data for at least 8 species. Therefore, the use of 3-species toxicity data could be promising for predicting the $HC_5$ values from which PNECs are frequently derived in ecological risk assessments. Note that our study was limited to log-normal SSDs based on acute toxicity data for 60 chemicals, whereas PNECs for long-term exposure to chemicals are derived from SSDs based on chronic toxicity data. However, because, for a large proportion of chemicals, collecting chronic toxicity data for three species would be much more difficult and less feasible than collecting acute toxicity data for three species (Hoondert et al., 2019; Posthuma et al., 2019), other prediction approaches, including the use of acute to chronic ratios for SSDs (De Zwart, 2002; Hiki & Iwasaki, 2020) and in silico predictive models (Dyer et al., 2008; Douziech et al., 2020; Takata et al., 2020), could be worth pursuing.

## ACKNOWLEDGEMENTS

We are grateful to Takehiko Hayashi, Ayako Furuhama, Kyoshiro Hiki, Masashi Kamo, and Wataru Naito for their useful comments and to Bin-Le Lin for her technical support. Comments from Takashi Nagai originally inspired the present analysis.

### Funding

This study was supported by a grant from the Japan Chemical Industry Association (JCIA) Long-range Research Initiative (LRI). The funders had no role in study design, data collection and analysis, decision to publish, or preparation of the manuscript.

## Grant Disclosures

The following grant information was disclosed by the authors:
Japan Chemical Industry Association (JCIA) Long-range Research Initiative (LRI).

## Competing Interests

The authors declare there are no competing interests.

## Author Contributions

- Yuichi Iwasaki conceived and designed the experiments, performed the experiments, analyzed the data, prepared figures and/or tables, authored or reviewed drafts of the paper, and approved the final draft.
- Kiyan Sorgog performed the experiments, analyzed the data, authored or reviewed drafts of the paper, and approved the final draft.

## Data Availability

All the data and R code used are available in the Supplemental Files.

## Supplemental Information

Supplemental information for this article can be found online at http://dx.doi.org/10.7717/peerj.10981#supplemental-information.

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
