# Peer review of "Estimating species sensitivity distributions on the basis of readily obtainable descriptors and toxicity data for three species of algae, crustaceans, and fish"

_PeerJ, doi:10.7717/peerj.10981_

## Round 0.1 · original submission · Major Revisions

The reviewer comments are self-explanatory and easy to follow. Please address or respond to them in your revision. One major comment involves comparing 3 vs. all species. Reviewers found it hard to understand what is compared where, in particular in the lower panels where models are based on both descriptors and 3 species data. Finally, I would encourage you to check the grammar on your revised manuscript before sending in the revision.

Reviewer 1 ·

Basic reporting

The paper "Estimating species sensitivity distributions on the basis of readily obtainable descriptors and toxicity data for three species of algae, crustaceans, and fish" presents linear regressions of HC5 and SDs vs Kow.
Introduction: The 1st and 2nd objectives cover all and three species respectively, more or less repeating the Hoondert and Douziech studies. The 3rd objective appears to be to compare the two but that is not clear from the wording. A scheme in the Methods might helpt to explain this.

Experimental design

Desk study

Validity of the findings

The comparisons between 3 and all species, with/without descriptors, are confusing and the tables and figures do not help:
Fig. 1 Caption incomplete text. Legend of different dots (chemical classes? not given. Wording confusing, "estimates based on measured" appears to be just "measured", and "estimates based on developed models" just "modelled". "Only descriptors" should be "Only descriptors mean / SD" to be consistent with the other panels
Fig 2. Now all dots are the same. Yet, for the interpretation is crucial the the MoA is known.
Table 2 I do not understand how to interpret the values, appear to be coefficients of a regression formula not given in the caption.

Additional comments

I can see the added value of comparing 3 vs. all species, based on data only and on regressions with descriptors. Yet, after reading an hour, I still find it hard to understand what is compared where, in particular in the lower panels where models are based on both descriptors and 3 species data, which appears rather unlikely to be applied in regulation. Hence, I suggest to illustrate the calculation by a scheme, simplify the results presented, use consistent legend etc. to get readers like me on board.

·

Basic reporting

Overall, the article is well written, clear, and easy to follow. Here and there some sentences are very long and could be shortened as suggested in the next comments. The argumentation in the introduction is sometimes not clear enough (LL44-57). The structure is adequate, with some bits that could be moved as suggested in the comments below. The figures and tables are relevant, the raw data supplied.

Abstract: very long sentence staring with “Here,” on L16-20 please shorten and rephrase. Also mention in the abstract that 45 chemicals were used for model fitting and 15 for validation.
Introduction: LL44.57: the line of thought is difficult to follow. I would suggest the author explain the findings in Hoondert et al 2019 first, highlighting the need for models predicting the mean and SD of SSDs separately and the difficulty of accurately modelling SDs using only chemical-specific data (given the argument on L54). The sentence on LL65-69 can be used here.
LL70-72 is too much focused on the methods to be included in the introduction.

Experimental design

The study presents a very interesting and promising approach to derive toxicological information on a chemical in case few data is available. However, the chemical dataset used by the authors is too small in my opinion to ensure a good coverage of the large variability of chemicals encountered. I ask for a major revision of the paper considering (1) the use of a more extended database like the one published in Posthuma et al 2019, (2) the consideration of using experimentally-derived physico-chemical properties instead of estimated ones.
(1) Posthuma et al., 2019 published an extensive database available upon request with more than 12 000 chemicals, which the authors already cite. I suggest the authors investigate this database and provide a clear explanation why they did not test their suggested approach on this extended dataset. In case such explanation cannot be found, I suggest the authors apply their methodology on this large set of data since it would greatly increase the relevance of their proposed method and its potential use for future research.
(2) Physico-chemical property predictions are not always accurate so I suggest the authors look for experimental values for the predictors used. This could increase the reliability of their model.
In addition, the following comments should be considered to make the methods clearer:
• L136: were the means and SDs of the toxicological data of three species calculated per chemical? And the 3000 bootstrapped samples were per chemical, a random combination of three species from algae, crustacean, and fish? I would suggest to rephrase this because this part is very difficult to understand.
• Did the authors ensure that during bootstrapping, the 3000 sample sets randomly drawn were not the same?
• LL176-178: belongs to the methods in my opinion
• Could Table 1 be extended to describe more the chemicals included. Potentially by classifying all chemicals into the Verhaar scheme? Or using the same classification as reported in Hoondert et al 2019.
• L156: were the models (3) fitted using each time (so for SD and mean) the mean and SD of the three species?

Validity of the findings

LL234-237: I do not agree with this statement: indeed it is difficult to clearly understand the driving factors, but Hoondert et al 2019 do account for the quality of the data (see Figure 1 of the paper) and also test the models on a much larger dataset used (around 300 data points).
L245-256: I understand the point the authors are trying to make here, but the argumentation is not clearly articulated in my opinion. I suggest a rephrasing starting with the sentence on LL249-253, and then explaining that Douziech et al 2020 showed a large uncertainty of estimated HC5 values based on three species because of the lack of data (sampling uncertainty, described in the SI Equations 4-6) and finishing with LL253-256. At this point, the authors could also mention the uncertainty in their model coming from the fact that the predictors relied on estimated values.
LL282-286: very long sentence, please shorten.
LL288-290: if the authors choose not to extend their database to more chemicals, I recommend to reflect on the number of datapoints used to derive the model in the conclusion too.

·

Basic reporting

Sufficient field background is provided. I suggest some additional references in General comments. The structure of article is fine. The unit of the paper is self-contained.

Experimental design

The content meets the Aims and Scope of PeerJ. Research question is well defined. I suggest Method section should be more detailed (See General comments).

Validity of the findings

All underlying data have been provided as Supplemental Data. Conclusions are well stated and linked to original research question.

Additional comments

This paper demonstrated the predictive ability of the species sensitivity distribution (SSD) using limited ecotoxicological data. SSD approach is highly useful for the derivation of water quality benchmarks and quantitative ecological risk assessment of chemical substances. However, the quality and quantity of available toxicity data varied markedly among substances, and therefore, estimation of SSD using limited data would promote the practical use of SSD.

In the present study, the predictive ability of SSD was improved using 3-species mean and SD of acute toxicity, as well as previous descriptors (physico-chemical properties and information on mode of action). The analysis is reasonable and well described in the paper. Therefore, I recommend accepting for PeerJ. Minor comments are listed below and should be addressed before being accepted.

1. line 97-99
Categorization of reliability into four classes complies with MANUAL FOR INVESTIGATION OF HPV CHEMICALS: CHAPTER 3 DATA EVALUATION by OECD and Klimisch HJ, Andreae E and Tillmann U (1997) A systematic approach for evaluating the quality of experimental and ecotoxicological data. Reg.Tox. Pharm. 25:1-5. The references should be provided.

2. line 106-110
The references of these criteria should be provided.

3. line 112-114
"the means and SDs" should be logarithmic means and logarithmic SDs. The description should be revised for accuracy throughout the paper. Moreover, the method of SSD analysis (there are multiple ways for determination of mean and SD) should be described. See "Aldenberg and Jaworska (2000) Ecotoxicol. Environ. Saf. 46:1-18".

4. line 114-116
This part should be moved to "Model evaluation (line 152-)".

5. line 119-124
When these descriptors are all estimated values (not measured values), the uncertainty of these estimations would affect the SSD estimation model. The description about this uncertainty should be included in Discussion part.

6. line 127-131
As with comment 5, the uncertainty in estimating mode-of-action classification should be described in Discussion part.

7. line 139
the three species -> each one species from three biological groups?

8. line 142-144
I couldn't figure out what the authors did. More detailed information is necessary.

9. line 265-271
I suggest it is helpful to indicate the difference in the predictive ability of each chemical class.

10. Figure2
More information on deriving 95% confidence intervals is necessary (preferably in Method section, not in the figure caption). Was parametric bootstrapping (n = 3000) applied to the best estimation model? It is confusing with line 138 (produced 3000 bootstrapped sample). Moreover, HC5 should be Log10 HC5 (if correct).

---

## Round 0.2 · Minor Revisions

Thanks for your efforts in improving your manuscript in response to reviewer comments. Please consider addressing Reviewer 2's minor comments on your revision.

Reviewer 1 ·

Basic reporting

Presuming that the authors are eager to get their message across clearly, I did not check improvement of wording of the revision.

Experimental design

Though simple, the new figure on the Methods is helpful to quickly confirm whether one's interpretation of the text is correct.

Validity of the findings

I now understand the results. I asked for figures expressing the MoA in the symbols but the authors choose to stick to the train vs test set subdivision. So, I turned to the table to learn that only 1 chemical with a specific MoA was included in the test set. As toxicities of specific MoAs (class 4) and to a somewhat lesser extend reactive MoAs (class 3) are most difficult to predict, the outcomes of the paper are largely restricted to (non-)polar narcosis (class 1-2). Hence, I suggest to include this in the conclusions and abstract. Nevertheless, showing that non-polar narcosis is better predicted by 3 species data rather than overall QSARs is still interesting.

·

Basic reporting

The authors carefully reviewed their manuscript and provided clear and justified answers to each point I raised in the first review. The manuscript now reads better and I believe that all the sentences are in their right section.

Experimental design

I understand the comment of the authors justifying their choice of a smaller database. In fact, the study deserves publication as is, and I appreciate the efforts to state more clearly and often on how many data points the model relies. I really hope the authors test the validity of their approach on a larger dataset in a future study.
I just spotted one mistake in my opinion on LL128 where the authors mention 31 SSDs with information for at least 10 species. Looking at the SI it should be 40, I believe. (Also to reach the 60 chemicals).

Validity of the findings

LL280-281: I still do not agree with this statement: “however, they did not take the quality of each of the toxicity values into consideration.” This is not true since in Figure 1 of Hoondert et al 2019 they do differentiate high and moderate quality data, they therefore excluded low quality data from the analysis. Please rephrase this statement.

Additional comments

Overall, after considering the two minor comments stated above, I believe that this manuscript is ready for publication.

·

Basic reporting

I evaluated in the first review.

Experimental design

I evaluated in the first review.

Validity of the findings

I evaluated in the first review.

Additional comments

I checked that my first review comments were properly addressed in the revised manuscript.

---

## Round 0.3 · accepted · Accept

Thank you for your hard work on your manuscript.